# Lifestyle Medicine and Psychological Well-Being Toward Health Promotion: A Cross-Sectional Study on Palermo (Southern Italy) Undergraduates

**DOI:** 10.3390/ijerph17155444

**Published:** 2020-07-28

**Authors:** Domenica Matranga, Vincenzo Restivo, Laura Maniscalco, Filippa Bono, Giuseppe Pizzo, Giuseppe Lanza, Valerio Gaglio, Walter Mazzucco, Silvana Miceli

**Affiliations:** 1Department of Health Promotion, Mother and Infant Care, Internal and Specialized Medicine “G. D’Alessandro”, University of Palermo, 90127 Palermo, Italy; domenica.matranga@unipa.it (D.M.); vincenzo.restivo@unipa.it (V.R.); giuseppeluigimarco.lanza@you.unipa.it (G.L.); valerio.gaglio@unipa.it (V.G.); walter.mazzucco@unipa.it (W.M.); 2Department of Biomedicine, Neuroscience and Advanced Diagnostics, University of Palermo, 90127 Palermo, Italy; 3Department of Economics, Business and Statistics, University of Palermo, 90128 Palermo, Italy; filippa.bono@unipa.it; 4Department of Surgical, Oncological and Oral Sciences, University of Palermo, 90127 Palermo, Italy; giuseppe.pizzo@unipa.it; 5Department of Psychology, Educational Science and Human Movement, University of Palermo, 90128 Palermo, Italy; silvana.miceli56@unipa.it

**Keywords:** lifestyle medicine, chronic diseases, modifiable behaviours, risk factors, psychological well-being, public health, eudaimonia, hedonia

## Abstract

(1) Aim: To assess the attitude toward Lifestyle Medicine and healthy behaviours among students in the healthcare area and to demonstrate its association to psychological well-being; (2) Methods: A cross-sectional study is conducted among 508 undergraduates of the University of Palermo (140 (27.6%) in the healthcare area and 368 (72.4%) in the non-healthcare area), during the academic year 2018–2019. Psychological well-being is measured through two dimensions of eudaimonia and hedonia, using the 10-item Hedonic and Eudaimonic Motives for Activities-Revised (HEMA-R) scale, with answers coded on a 7-point scale. The association between demographic and modifiable behavioural risk factors for chronic diseases is assessed through crude and adjusted Odds ratios with 95% confidence intervals; (3) Results: Orientation to both hedonia and eudaimonia is significantly associated to the Mediterranean diet (ORAdj = 2.28; 95% CI = (1.42–3.70)) and drinking spirits less than once a week (ORAdj = 1.89; 95% CI = (1.10–3.27)) and once a week or more (ORAdj = 6.02; 95% CI = (1.05–34.52)), while these conditions occur together less frequently for current smokers (ORAdj = 0.38; 95% CI = (0.18–0.81)). Students inclined to well-being consider healthcare professionals as models for their patients and all people in general (OR = 1.96, 95% CI = (1.28–3.00)); (4) Conclusions: The positive relation found between a virtuous lifestyle and psychological well-being suggests the construction, development and cultivation of individual skills are a means to succeed in counteracting at risk behaviours for health.

## 1. Introduction

Worldwide, countries put policy strategies and health programs in place to counteract the spread chronic diseases and to promote healthy behaviours [1]. Poor diet, physical inactivity, tobacco, and alcohol abuse are considered the modifiable behavioural risk factors (MBRF) determining the greatest burden of chronic diseases [2]. The World Health Organization [1] reports that tobacco accounts for over 7.2 million deaths every year, including those related to being exposed to passive smoke, and is expected to increase markedly over the coming years. It also estimates that excess salt/sodium intake is responsible for 4.1 million annual deaths and that insufficient physical activity accounts for 1.6 million deaths annually. Further, nearly 3 million annual deaths are attributable to alcohol abuse (5.3% of all deaths) [1]. To support the change of lifestyle as a first line of prevention, the Lifestyle Medicine (LM) approach, as defined in 2010 by the Journal of the American Medical Association, is “the evidence-based practice of assisting individuals and families to adopt and sustain behaviours that can improve health and quality of life” [3]. The American College of Lifestyle Medicine (ACLM) establishes six ways to take control of health through regular physical exercise, adequate and good quality sleep, cessation of smoking, stress management and maintenance of relationships. Moreover, LM advocates for a whole-food, plant-based diet for a nutritional lifestyle that is based on whole, minimally processed foods, plants, including vegetables, fruits, whole grains, legumes, seeds and nuts, and little or no intake of animal products. Additionally, there is a preference for quality or organic food and zero-miles products. Physicians are the first source of information about health and healthy behaviours for their patients [4] and physician counselling has been demonstrated to be effective for weight loss [5], smoke cessation [6] and alcohol intake [7]. Regarding the skills required to practice LM, primary care physicians are called to promote healthy behaviours as the basis of medical care, disease prevention and health promotion, as well as providing the knowledge of the mechanisms by which specific lifestyle changes can have a positive effect on patient health outcomes [8]. Considering this, it is essential to involve medical students in shaping and acquiring these skills through education and the alignment of medical curricula. Medical students report a lack of knowledge and poor skills to counsel their patients to improve lifestyle behaviours [9]. Even if the need to steer medical students to disease prevention and health promotion has been acknowledged, it must be noted that medical curricula are devoid of some content, such as positive psychology and psychological well-being [10], which are needed to perform counselling effectively [11].

Individuals experiencing a state of well-being are less likely to report alcohol and drug addiction, be on a poor diet or be sedentary [12], and to develop and maintain better social relationships [13]. Recently, the interest in well-being within the psychological field has grown rapidly to investigate the sources of happiness and the many facets of human flourishing [14]. Taking this perspective, we can distinguish two different approaches: “hedonia” that focuses on pleasure, happiness and the achievement of well-being through the satisfaction of one’s desires [15] and “eudaimonia”, according to which well-being is obtained by fulfilling one’s potential in the pursuit of complex and meaningful goals [16]. According to Huta [17], the eudaimonic orientation is defined in terms of four core elements: authenticity, meaning, excellence and growth. The first refers to the possibility of living and acting in accordance with one’s values and one’s self; the second concerns the ability to identify the meaning of things and their value; the third refers to the struggle of the human being to reach high standards of ethical behaviour; and the fourth, finally, concerns the ability to actualise one’s potential. Consequently, the eudaimonic vision is based on the concept of flourishing, true self, actualising potential, meaning, objectives, purposes, and personal expression and implies a process of continuous construction, development, and cultivation of individual skills. To contrast, the author defines the hedonic orientation as the pursuit of two elements: pleasure, that is the search for pleasant sensations and emotions; and comfort, the tendency to seek easy and painless solutions. A hedonic orientation, therefore, pushes the individual in search of what is subjectively pleasant [17].

Taking the perspective of medical education, the positive relation between a virtuous lifestyle and psychological well-being suggests promoting the development of curricula where a varied spectrum of LM topics, such as nutrition and exercise together with the analysis of the underlying psychological processes, are provided and developed in an integrated manner [18].

To assess the attitude toward LM and lifestyle behaviours among students in the healthcare area, and to demonstrate the association with psychological well-being, we conduct a cross-sectional study among undergraduates of the University of Palermo during the academic year 2018–2019. More in depth, our study assumes that the adoption of healthy behaviours could be significantly associated with psychological well-being.

## 2. Materials and Methods

### 2.1. Study Design

A cross-sectional study was designed to investigate the study aims on a sample of students attending health (medicine, dentistry, and other health professional courses) and non-health (economics and psychological science) degree courses. To this end, a questionnaire articulated into five sections (demographics, diet habits, smoking, physical activity, and psychological well-being) was structured. To assess the lifestyle, composite indicators were constructed for (i) eating habits, (ii) alcohol consumption, (iii) sedentary lifestyle, (iv) smoking, (v) sexual habits, and (vi) addictions. The diet habits section included questions regarding variations which occurred in the last two and five years in the respondent’s dietary habits, the food frequency test, and questions about alcohol intake [19], and binge drinking. The most widespread healthy diet in Italy is the Mediterranean diet that consists of “a high intake of vegetables, legumes, fruits and nuts, and cereals, and a high intake of olive oil but a low intake of saturated lipids, a moderately high intake of fish, a low-to-moderate intake of dairy products, a low intake of meat and poultry, and a regular but moderate intake of ethanol, primarily in the form of wine and generally during meals” [20]. Following Trichopoulou et al. [19], for subjects whose consumption was adherent to the Mediterranean diet, a score from 0 (minimum adherence) to 10 (maximum adherence) was assigned for each respondent and the median value was chosen as a cut-off to define the respondent’s adherence to the Mediterranean diet (adherent if the score > median and not adherent if the score ≤ median). Alcohol consumption was assessed concerning wine, beer and spirits and expressed through three frequency categories (never, less than once a week, once a week or more).

Binge drinking typically happens when men consume 5 or more drinks or women consume 4 or more drinks in about 2 hours. Binging was detected through a question about the occurrence of a binge episode in the last 12 months and expressed as a binary variable with “yes” or “no” response categories [21].

The smoking section was developed in agreement with the WHO guidelines about tobacco use. Smoking status was expressed as a categorical variable with three categories, non-smoker, former and current smoker [1]. Physical activity was defined as a binary variable with “yes” or “no” as the response categories [22].

The measure used to assess eudaimonic and hedonic orientations is the Hedonic and Eudaimonic Motives for Activities-Revised (HEMA-R) scale, which represents a revision of the original Hema scale [23]. The scale is made up of 10 items, five of which refer to eudaimonic orientation and five to hedonic orientation. Particularly, hedonia is expressed through the concepts of “pleasure” and “comfort”. Conversely, eudaimonia is demonstrated through the concepts of “authenticity”, “excellence”, “growth” and “meaning”. The subjects were asked to evaluate the degree to which they usually carry out their activities according to the intentions indicated and independently of the achievement of the objective. Participants gave ratings on various eudaimonic and hedonic motives, which were intermixed on a scale from 1 (not at all) to 7 (very much). A subject was considered oriented to hedonia or eudaimonia if the respective score was not less than the median. Well-being was expressed through a categorical variable with four categories (no orientation, hedoniac, eudaimoniac, both orientations).

To assess each student’s health, each subject was asked if he/she had ever suffered from diabetes, hypertension, hypercholesterolemia, other cardiovascular diseases, neoplasia, or other chronic diseases. The occurrence of chronic disease was expressed as a binary variable with category “yes” in the case of at least one chronic disease and “no” in the case of none.

Finally, the questionnaire included three specific questions about the role of healthcare professionals for prevention and health promotion. First, each participant was asked if healthcare professionals could be considered as models for their patients and people in general; second, if they should regularly counsel their patients for tobacco cessation and, third, if a patient has an increased probability to stop smoking if assisted by his/her practitioner or nurse. Regarding all questions, the possible answer categories were “yes” and “no”.

The study was presented by the research team during a seminar entitled “Communicative skills for the promotion of correct lifestyles and the prevention of chronic degenerative diseases” opened to students from different study areas. Then, students were invited to adhere to the survey by self-administering the questionnaire through the Google platform. The study was approved by the Ethical Committee of the “Azienda Universitaria Policlinico Paolo Giaccone” of Palermo (Reference number 08/2018).

### 2.2. Statistical Methods

Categorical variables were expressed as counts and percentages, and continuous variables were categorized using the median as the cut-off value. Univariable association between demographic and Modifiable Behavioural Risk Factors (MBRFs) was assessed using the Chi-square test or the Fisher exact test, as appropriate, and measured through crude Odds Ratios (OR) and 95% Confidence Intervals (CIs). Multinomial logistic regression was performed to assess the likelihood of being oriented toward eudaimonia, hedonia or both, compared to no orientation in relation to demographics and MBRFs that had significant results using univariable analysis. Results of multivariable analysis were expressed as adjusted ORs (AdjOR) and 95% CIs.

Stata IC/15.1 (StataCorp LLC, Texas, TX, USA) was used for statistical analysis, and a *p*-value < 0.05 was chosen as the statistical significance cut-off.

## 3. Results

A sample of 508 undergraduate students, 140 (27.6%) from the health area and 368 (72.4%) from the non-health area, was recruited during the study. The majority of the respondents attended the first year of study (n = 348; 68.5%), while the remaining part was divided between the second and third attendance year, 81 (15.9%) and 79 (15.6%), respectively.

The Hedonic and Eudaimonic Motives for Activities-Revised (HEMA-R) was available for all students, 325 (64.0%) females, with parents highly educated or graduated (79% fathers and 81% mothers), mostly on-site students (267, 43.8%) followed by off-site (142, 28%) and commuter (99, 15.5%) students. The attitude to be hedonic was lower for females than for males (Odds Ratios (OR) = 0.56, 95% Confidence Intervals (CI) = (0.35–0.91)). Notably, there was not any significant difference between students in the healthcare and non-healthcare area to be more inclined to either hedonia or eudaimonia (Table 1).

Using univariable analysis, students following the Mediterranean diet were more inclined to eudaimonia (OR = 2.50, 95% CI = [1.52–4.12]), while those with binge experience in the past were less prone (OR = 0.48, 95% CI = [0.29–0.79]). Overweight students were more prone to hedonia than normal weight ones (OR = 2.33, 95% CI = [1.11–4.95]), as well as students drinking beer less than once a week (OR = 2.15, 95% CI = [1.14–4.05]) and once a week or more (OR = 2.81, 95% CI = [1.23–6.42]), and for students reporting taking spirits sometimes (OR = 2.01, 95% CI = [1.17–3.46]). Furthermore, there was a statistically significant association with experiencing binge drinking (OR = 1.62 95% CI = [1.00–2.62]). Students accustomed to tobacco were less likely to be oriented to both hedonia and eudaimonia (OR = 0.47 95% CI = [0.28–0.79] for former smokers and OR = 0.40 95% CI = [0.22–0.75] for current smokers). Conversely, students drinking spirits less than once a week (OR = 1.50 95% CI = [1.00–2.24]) and once a week or more (OR = 4.29 95% CI = [1.14–16.17]) and those reporting binging experiences (OR = 1.43 95% CI = [0.98–2.09]) were more oriented to both dimensions of psychological well-being (Table 2). 

Students inclined to both well–being dimensions considered healthcare professionals as models for their patients and all people in general (OR = 1.96, 95% CI = [1.28–3.00]). Students in the healthcare area were more inclined than those in non–healthcare fields to consider that healthcare professionals should counsel their patients for tobacco cessation on a regular basis (OR = 8.03; 95% CI = [1.05–61.3]) (Table 3). 

Using multivariable analysis, the Mediterranean diet was confirmed as the only one statistically significant for being eudaimonic (AdjOR = 3.27; 96% CI = [1.83–5.87]). Additionally, experiencing both hedonia and eudaimonia was significantly associated to the Mediterranean diet (AdjOR = 2.28; 95% CI = [1.42–3.70]) as well as drinking spirits less than once a week (AdORj = 1.89; 95% CI = [1.10–3.27]), while these conditions occurred together less frequently for current smokers (AdjOR = 0.38; 95% CI = [0.18–0.87]) (Table 4). 

## 4. Discussion

Lifestyle Medicine has produced significant changes in the concept of health, moving from a care–centred approach to an approach focused on promoting well–being. Our findings document a positive relationship between healthy behaviours and psychological well–being and suggest the construction, development, and cultivation of individual skills as a means to be successful in counteracting behaviours at risk for health. Specifically, we have deepened the link between eudaimonic well–being and healthy behaviours, with particular regard to nutrition, physical activity, alcohol, and tobacco abuse. Our results indicate that students adhering to the Mediterranean diet were more likely to be oriented toward psychological well–being, in line with other literature [12]. Additionally, taking alcohol less than once a week or once a week or more was significantly related to both hedonic and eudaimonic well–being. This surprising result may be explained as the occasional intake of alcohol in the young population typically occurs between meals within a social context characterized by interactions that strengthen individual identity and group cohesion [24]. Conversely, the result that current smokers are less oriented toward the hedonic and eudaimonic perspective can be explained as tobacco consumption is perceived as an addiction more than a condition of pleasure [25]. 

Different from the classical approaches followed by studies conducted on medical students and residents [26], our study explicitly examined the role of the eudaimonic dimension in influencing the implementation of healthy lifestyles. The results are in line with the most recent developments in the literature, which have shown that the pursuit of eudaimonic objectives undoubtedly represents a protective factor from the implementation of health risk behaviours [27].

The result that eudaimonic students showed a trend toward improvement in their own healthy behaviours is worthy of note for the effective education of future Lifestyle Medicine (LM) practitioners, as it has been demonstrated that there is a positive association between preventive healthy habits of physicians and their patients [28]. Eudaimonic subjects were found to be more prone to consider healthcare professionals as a model for the whole community and students in the healthcare field were more oriented to counselling than non–healthcare students.

Ryff [29] suggests the science of eudaimonia shows its relevance for multiple aspects of physical functioning, as well as for health research. Eudaimonic well–being is capable of leading the individual to functional and highly adaptive behaviour patterns. The assumption of a eudaimonic perspective that focuses on the subject could indirectly lead to a change in lifestyle. Unlike many coaching programs developed for the promotion of health that mainly aim to modify specific dysfunctional behaviours, other studies [30] have developed interventions whose participation induced greater subjective vitality in the subjects, leading them to develop changes in their lifestyles not explicitly requested. This is the typical approach of the LM practitioner, to steer the patient to commit oneself to achieve greater awareness of oneself and of one’s life meanings, to implement virtuous and healthy behaviours. The LM practitioner has the skills to assess patient attitudes toward making healthy behaviour changes, helping them in self–managing healthy behaviours and recognising negative stress responses, also with the aid of technology and multidisciplinary teamwork. Focusing on the eudaimonic perspective, therefore, means creating a setting in which to explore oneself, understand one’s potential, one’s sense of life, far from the emotional stress derived from programs based on diets, exercise, or reduction in tobacco consumption, as examples. Promoting well–being, in any of its forms, therefore, can represent a useful goal to create conditions in which life is perceived as interesting, rewarding, and full of meaning. 

The study did not find any significant difference between students attending courses in the healthcare and non–healthcare areas concerning the relation between lifestyle and well–being. This suggests that enhancing the awareness of one’s well–being is useful for the balanced growth of all individuals, but it becomes an essential skill of valued students in the healthcare field and of effective LM practitioners. Therefore, our study encourages the development of curricula for medical students that include, in addition to topics such as nutrition and exercise, the analysis of the psychological processes underlying well–being and positive mental health. This result is in line with the most recent contents of the Lifestyle Medicine Core Competencies Program (https://lifestylemedicine.org/ACLM/Education/Certification). This program adds to core competency skills, nutrition, tobacco cessation, physical activity, sleep education, alcohol use and weight management, as well as the skills related to health and wellness coaching, emotional wellness, and mindfulness. The usual strategies used by the different countries to reduce alcohol and tobacco consumption or the traditional intervention programs aimed at pushing the population toward healthy eating behaviours have not always proved effective in the long term. 

There are some flaws in this study. The first one regards the recruitment of subjects. Actually, the two student groups were different as only those in the healthcare field were compelled to the seminar where the study was presented. Therefore, it can be hypothesised that students in the non–healthcare area was disproportionately motivated to partake in the research as they are the most interested and active students, who regularly follow seminars voluntarily. This may partly explain the absence of differences between the two groups of students. Another limitation regards the lack of some important information. Actually, anthropometric data of weight and height are self–reported and not measured, leading to under–reporting of weight while height was not indicated at all by the majority of the enrolled students. It is why the Body Mass Index was not included in our results. Analogously, information regarding family income, even if requested, was not supplied by respondents. Consequently, we could not assess the role of economic well–being in the relation between psychological well–being and the student’s healthy behaviours. Taking the view of LM, further research is needed to investigate if the found positive association between healthy nutrition and the eudaimonic approach is still valid when considering the whole-food, plant-based diet.

Beyond these limits, this study contributes to LM research, as it suggests that healthy behaviour is easier to be acquired and maintained by individuals who aspire to achieve their life potential, and proposes the endowment of students in the healthcare area with the skills of counselling and well–being for a more effective spread of correct lifestyles in the general population. The protective relationship between positive psychological functioning and physical health was found, thus, promoting correct lifestyles can have an impact on health also through the enhancement of psychological well–being.

## 5. Conclusions

The main finding of our study was the positive relationship between virtuous lifestyle behaviours and psychological well–being. Considering terms of health promotion, the enhancement of correct lifestyles could lead to the identification of new methods aimed at preventing noncommunicable diseases and reducing the prevalence of behaviours at risk for health. The eudaimonic perspective and the Lifestyle Medicine approach imply a change of route for general practice and health promotion, from an approach based on deprivation and bans to an approach based on patient empowerment and self–management of their own lifestyle behaviours.

## Figures and Tables

**Table 1 ijerph-17-05444-t001:** Psychological well-being by socio-demographic and health characteristics in a sample of 508 undergraduates of University of Palermo: univariable analysis.

Characteristics ^§^		Eudaimonia			Hedonia			Both		
	N	n (%)	OR (95%CI) ^§§^	*p-*Value	n (%)	OR (95%CI) ^§§^	*p*-Value	n (%)	OR (95%CI) ^§§^	*p*-Value
age (ys)				0.639			0.425			0.171
≤20	340	54 (15.88)	1.00		58 (17.06)			101 (29.71)	1	
>20	168	24 (14.29)	0.88 (0.52–1.49)		24 (14.29)	0.81 (0.48–1.36)		60 (35.71)	1.31 (0.89–1.95)	
gender				0.192			**0.018**			0.321
male	183	23 (12.57)	1		39 (21.31)	1		63 (34.43)	1.00	
female	325	55 (16.92)	1.42 (0.84–2.40)		43 (13.23)	0.56 (0.35–0.91)		98 (30.15)	0.82 (0.56–1.21)	
chronic diseases				0.102			0.243			0.462
no	491	73 (14.87)	1		81 (16.50)	1		157 (31.98)	1	
yes	17	55 (29.41)	2.39 (0.81–7.00)		1 (5.88)	0.32 (0.04–2.43)		4 (23.53)	0.65 (0.21–2.04)	
year of study										
first	348	55 (15.80)	1		56 (16.09)	1		109 (31.32)	1	
second	81	9 (11.11)	0.67 (0.31–1.40)	0.286	15 (18.52)	1.19 (0.63–2.23)	0.597	29 (35.80)	1.22 (0.74–2.03)	0.437
third	79	14 (17.72)	1.15 (0.60–2.19)	0.676	11 (13.92)	0.84 (0.42–1.70)	0.633	23 (29.11)	0.90 (0.53–1.54)	0.702
father education										
primary	106	14 (13.21)	1		14 (13.21)	1		35 (33.02)	1	
high	203	34 (16.75)	1.32 (0.67–2.59)	0.415	36 (17.73)	1.42 (0.72–2.77)	0.306	57 (28.08)	0.79 (0.48–1.32)	0.368
graduate or higher	198	30 (15.15)	1.17 (0.59–2.33)	0.647	32 (16.16)	1.27 (0.64–2.50)	0.494	68 (34.34)	1.06 (0.64–1.75)	0.816
mother education										
primary	96	13 (13.54)	1		16 (16.67)	1		34 (35.02)	1	
high	237	32 (13.50)	1.00 (0.50–2.00)	0.992	38 (16.03)	0.95 (0.50–1.81)	0.887	74 (31.22)	0.83 (0.50–1.37)	0.460
graduate or higher	174	33 (18.97)	1.49 (0.74–3.01)	0.257	28 (16.09)	0.96 (0.49–1.88)	0.903	52 (29.89)	0.78 (0.46–1.32)	0.351
type of student residency										
on-site	267	33 (12.36)	1		51 (19.1)	1		84 (31.46)	1	
off-site	142	28 (19.72)	**1.74 (1.00–3.03)**	**0.047**	19 (13.39)	0.65 (0.37–1.16)	0.144	48 (33.80)	1.11 (0.72–1.72)	0.630
commuter	99	17 (17.17)	1.47 (0.78–2.79)	0.234	12 (12.12)	0.58 (0.30–1.15)	0.117	29 (29.29)	0.90 (0.54–1.50)	0.691
type of course				0.215			0.914			0.770
non-healthcare area	368	52 (14.13)	1		59 (16.03)	1		118 (32.07)	1	
healthcare area	140	26 (18.57)	1.39 (0.83–2.33)		23 (16.43)	1.03 (0.61–1.74)		43 (30.71)	0.94 (0.62–1.43)	

**^§^** Statistics were calculated on available data; Bold values are statistically significant; ^§§^ OR: Odds ratio; CI: Confidence Interval.

**Table 2 ijerph-17-05444-t002:** Psychological well-being by behavioral characteristics in a sample of 508 undergraduates of University of Palermo: univariable analysis.

Characteristics ^§^		Eudaimonia			Hedonia			Both		
	N	n (%)	OR (95%CI) ^§§^	*p-*Value	n (%)	OR (95%CI) ^§§^	*p-*Value	n (%)	OR (95%CI) ^§§^	*p-*Value
in the last 5 years, did you change your weight?				0.428			0.231			0.403
no	119	21 (17.65)	1		15 (12.61)	1		34 (28.57)	1	
yes	389	57 (14.65)	0.80 (0.46–1.39)		67 (17.22)	1.44 (0.79–2.64)		127 (32.65)	1.21 (0.77–1.90)	
in this month, are you on a diet?				0.682			0.881			0.712
no	387	58 (14.99)	1		63 (16.28)	1		121 (31.27)	1	
yes	121	20 (16.53)	1.12 (0.64–1.96)		19 (15.70)	0.96 (0.55–1.68)		40 (33.06)	1.09 (0.70–1.68)	
adherence to Mediterranean diet ^2^				**<0.001**			0.241			0.060
no	305	32 (10.49)	1.00		54 (17.7)	1		87 (28.52)	1.	
yes	203	46 (22.66)	**2.50 (1.52–4.12)**		28 (13.79)	0.74 (0.45–1.22)		74 (36.45)	1.44 (0.98–2.10)	
weight status										
normal	262	51 (19.47)	1.00		37 (14.12)	1		87 (33.21)	1	
overweight	47	7 (14.89)	0.72 (0.31–1.71)	0.461	13 (27.66)	**2.33 (1.11–4.95)**	**0.021**	14 (29.79)	0.85 (0.43–1.68)	0.646
obese	50	7 (14.00)	0.67 (0.29–1.59)	0.363	5 (10.00)	0.68 (0.25–1.82)	0.435	18 (36.00)	1.13 (0.60–2.13)	0.702
smoking status										
non–smoker	76	10 (13.16)	1		10 (13.16)	1		36 (47.37)	1	
former smoker	312	57 (18.27)	1.48 (0.71–3.05)	0.291	47 (15.06)	1.17 (0.56–2.44)	0.674	93 (29.81)	**0.47 (0.28–0.79)**	**0.004**
current smoker	120	11 (9.17)	0.67 (0.27–1.66)	0.380	25 (20.83)	1.74 (0.78–3.88)	0.173	32 (26.67)	**0.40 (0.22–0.75)**	**0.003**
physical activity				0.514			0.290			0.154
no	219	31 (14.16)	1		31 (14.16)	1		62 (28.31)	1	
yes	289	47 (16.26)	1.18 (0.72–1.93)		51 (17.65)	1.30 (0.80–2.11)		99 (34.26)	1.32 (0.90–1.93)	
drinking wine										
never	153	26 (16.99)	1		20 (13.07)	1		44 (28.76)	1	
less than once a week	278	40 (14.39)	0.82 (0.48–1.41)	0.473	45 (16.19)	1.28 (0.73–2.27)	0.388	91 (32.73)	1.21 (0.78–1.86)	0.395
once a week or more	67	11 (16.42)	0.96 (0.44–2.08)	0.917	15 (22.39)	1.92 (0.91–4.06)	0.083	23 (34.33)	1.29 (0.70–2.40)	0.410
drinking beer										
never		26 (17.45)	1		14 (9.40)	1		43 (28.86)	1	
less than once a week	149	45 (15.79)	0.89 (0.52–1.51)	0.657	52 (18.25)	**2.15 (1.14–4.05))**	**0.015**	91 (31.93)	1.16 (0.75–1.78)	0.511
once a week or more	285	7 (11.29)	0.60 (0.25–1.48)	0.263	14 (22.58)	**2.81 (1.23–6.42)**	**0.010**	21 (33.87)	1.26 (0.67–2.39)	0.472
drinking spirits										
never	193	34 (17.62)	1		21 (10.88)	1		50 (25.91)	1	
less than once a week	294	42 (14.29)	0.78 (0.48–1.28)	0.322	58 (19.73)	**2.01 (1.17–3.46)**	**0.010**	101 (34.35)	**1.50 (1.00–2.24)**	**0.049**
once a week or more	10	1 (10.00)	0.52 (0.06–4.27)	0.535	1 (10.00)	0.91 (0.11–7.58)	0.931	6 (60.00)	**4.29 (1.14–16.17)**	**0.019**
binging										
no	255	51 (20.0)	1		33 (12.94)	1		71 (27.84)	1	
yes	253	27 (10.7)	**0.48 (0.29–0.79)**	**0.004**	49 (19.37)	**1.62 (1.00–2.62)**	**0.049**	90 (35.57)	**1.43 (0.98–2.09)**	**0.061**

**^§^** Statistics were calculated on available data. Bold values are statistically significant; ^§§^ OR: Odds ratio; CI: Confidence Interval.

**Table 3 ijerph-17-05444-t003:** Role of healthcare professionals for prevention and health promotion using a well–being approach and type of course in a sample of 508 undergraduates of University of Palermo.

Characteristics ^§^	Do You Think that Healthcare Professionals Could Be Considered as Models for Their Patients and People in General?			Do You Think that Healthcare Professionals Should Counsel Their Patients for Tobacco Cessation on a Regular Basis?			Do You Think that a Patient Has an Increased Probability to Stop Smoking, if Assisted by His/Her Practitioner or Nurse?		
	N	n (%)	OR (95%CI) ^§§^ *p*-Value	N	n (%)	OR (95%CI) ^§§^ *p*-Value	N	n (%)	OR (95%CI) ^§§^ *p*-Value
eudaimonia									
no	172	21 (12.21)		21	3 (14.29)		88	9 (10.23)	
yes	334	57 (17.07)	1.48 (0.86–2.54) 0.152	485	75 (15.46)	1.10 (0.32–3.82) 0.884	419	69 (16.47)	1.73 (0.83–3.62) 0.141
hedonia									
no	172	34 (19.77)		21	3 (14.29)		88	17 (19.32)	
yes	334	47 (14.07)	0.66 (0.41–1.08) 0.098	485	79 (16.29)	1.17 (0.34–4.06) 0.808	419	65 (15.51)	0.77 (0.42–1.39) 0.379
both									
no	172	39 (22.67)		21	9 (42.86)		88	29 (32.95)	
yes	334	122 (36.53)	**1.96 (1.28–3.00)** **0.002**	485	151 (31.13)	0.60 (0.255–1.46) 0.259	419	132 (31.50)	0.94 (0.57–1.53) 0.791
type of course									
non–healthcare	366	238 (65.0)		366	346 (94.5)		367	296 (80.7)	
healthcare	140	96 (68.6)	1.17 (0.77–1.78) 0.451	140	139 (99.3)	**8.03 (1.05–61.3)** **0.013**	140	123 (87.9)	1.74 (0.98–3.08) 0.056

**^§^** Statistics were calculated on available data; Bold values are statistically significant; ^§§^ OR: Odds ratio; CI: Confidence Interval.

**Table 4 ijerph-17-05444-t004:** Multivariable analysis of well–being by modifiable behavioral characteristics in a sample of 508 undergraduates of University of Palermo.

Characteristics^§^	Eudaimonia		Hedonia		Both	
	AdjOR (95% CI) ^§§^	*p*-Value	AdjOR (95% CI) ^§§^	*p*-Value	AdjOR (95% CI) ^§§^	*p*-Value
gender						
female	0.92 (0.49–1.71)	0.788	0.60 (0.34–1.07)	0.086	0.71 (0.44–1.15)	0.164
adherence to Mediterranean diet						
yes	**3.27 (1.83–5.87)**	**<0.001**	1.45 (0.80–2.63)	0.221	**2.28 (1.42–3.70)**	**0.001**
smoking status						
former smoker	1.50 (0.61–3.70)	0.375	1.23 (0.51–2.95)	0.645	0.63 (0.32–1.21)	0.163
current smoker	0.60 (0.20–1.77)	0.356	1.03 (0.41–2.62)	0.950	**0.38 (0.18–0.81)**	**0.012**
drinking wine						
less than once a week	0.84 (0.43–1.66)	0.621	0.86 (0.43–1.72)	0.674	0.89 (0.51–1.56)	0.695
once a week or more	1.35 (0.47–3.86)	0.575	1.54 (0.56–4.23)	0.406	1.07 (0.44–2.59)	0.878
drinking beer						
less than once a week	1.42 (0.72–2.81)	0.307	2.08 (0.98–4.42)	0.057	1.23 (0.70–2.16)	0.473
once a week or more	1.44 (0.43–4.86)	0.553	1.88 (0.62–5.71)	0.264	1.13 (0.45–2.83)	0.788
drinking spirits						
less than once a week	1.56 (0.81–2.99)	0.185	1.98 (1.00–3.89)	0.050	**1.89 (1.10–3.27)**	**0.021**
once a week or more	2.68 (0.21–34.33)	0.449	1.83 (0.15–23.06)	0.640	**6.02 (1.05–34.52)**	**0.044**
binging						
yes	0.57 (0.29–1.11)	0.096	1.11 (0.58–2.09)	0.758	1.11 (0.65–1.87)	0.707

^§^ Reference categories: “No” for Adherence to Mediterranean Diet, Physical activity, Binging, “Male” for Gender, “Non–smoker” for Smoking status, “Never” for Drinking wine, Drinking beer, Drinking spirits. Statistics were calculated on available data; Bold values are statistically significant; ^§§^ AdjOR: Adjusted Odds ratio; CI: Confidence Interval.

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
