# Peer review of "Lifestyle Medicine and Psychological Well-Being toward Health Promotion: A Cross-Sectional Study on Palermo (Southern Italy) Undergraduates"

_ijerph, 2020, doi:10.3390/ijerph17155444_

Round 1

Reviewer 1 Report

This is an outstanding paper. I believe that readers from the U.S. may not be familiar with the terms eudaimonia and hedonia so possibly explaining these or putting a brief description in parenthesis early on would be helpful. Also, I would like to see more of a focus on what was included in a Mediterranean diet or a focus in the discussion about whether the benefit was due to the whole food, plant based/predominant, limited processed food type of eating. Also, the American College of Lifestyle Medicine is mentioned which I appreciate but it maybe be worth noting that they advocate for a whole food, plant predominant way of eating as being a "north star" to follow. I don't think that a conflict of interest exists but it should be noted that I'm the current secretary of the American College of Lifestyle Medicine. Their research was very interesting and I appreciated the opportunity to review it. Thanks!

Author Response

Reviewer 1:

This is an outstanding paper. I believe that readers from the U.S. may not be familiar with the terms eudaimonia and hedonia so possibly explaining these or putting a brief description in parenthesis early on would be helpful. Also, I would like to see more of a focus on what was included in a Mediterranean diet or a focus in the discussion about whether the benefit was due to the whole food, plant based/predominant, limited processed food type of eating. Also, the American College of Lifestyle Medicine is mentioned which I appreciate but it maybe be worth noting that they advocate for a whole food, plant predominant way of eating as being a "north star" to follow. I don't think that a conflict of interest exists but it should be noted that I'm the current secretary of the American College of Lifestyle Medicine. Their research was very interesting and I appreciated the opportunity to review it. Thanks!

General reply to Reviewer 1:

We wish to thank you all for your constructive comments in this round of review. Your comments provided valuable insights to refine its contents and analysis. In this document, we try to address the issues raised as best as possible.

Comment 1:

I believe that readers from the U.S. may not be familiar with the terms eudaimonia and hedonia so possibly explaining these or putting a brief description in parenthesis early on would be helpful.

Reply to Reviewer 1:

Related to the description of the terms eudaimonia and hedonia we added the following statements in the paper: “According to Huda, the eudaimonic orientation is defined in terms of four core elements: authenticity, meaning, excellence and growth. The first refers to the possibility of living and acting in accordance with one’s values and one’s self; the second concerns the ability to identify the meaning of things, their value; the third, refers to the struggle of the human being to reach high standards of ethical behaviour; and the fourth, finally, concerns the ability to actualize one’s potential.” and “In contrast, the author defines the hedonic orientation as the pursuit of two elements: pleasure that is, the search for pleasant sensations and emotions; and comfort, i.e. the tendency to seek easy and painless solutions. A hedonic orientation, therefore, pushes the individual in search of what is subjectively pleasant”.

Comment 2:

Also, I would like to see more of a focus on what was included in a Mediterranean diet or a focus in the discussion about whether the benefit was due to the whole food, plant based/predominant, limited processed food type of eating. Also, the American College of Lifestyle Medicine is mentioned which I appreciate but it maybe be worth noting that they advocate for a whole food, plant predominant way of eating as being a "north star" to follow.

Reply to comment 2:

Related to what is included in a Mediterranean diet we added the following statement in the text “a high intake of vegetables, legumes, fruits and nuts, and cereals, and a high intake of olive oil but a low intake of saturated lipids, a moderately high intake of fish, a low-to-moderate intake of dairy products, a low intake of meat and poultry, and a regular but moderate intake of ethanol, primarily in the form of wine and generally during meals”, with a new reference: Willett WC, Sacks F, Trichopoulou A, et al. Mediterranean diet pyramid: a cultural model for healthy eating. Am J Clin Nutr 1995;61:Suppl 6:S1402-S1406

Reviewer 2 Report

Thank you for your work on this study.  I read the manuscript with interest and enthusiasm.  I especially appreciate the study design and analysis.  I would suggest the authors consider the following:

  1. The Aim is stated as to a. assess the attitude towards LM b. assess the attitude towards healthy lifestyle behaviors c. demonstrate an association with psychological wellbeing. It is unclear to me that these three Aims were clearly discussed in the paper and conclusion.  It is also unclear if these are the three items that the paper purports to address.
  2. There is language in the paper that are sweeping generalizations and unsupported and are not generalizable.  For example, the paragraph on page 8 that starts "Beyond these limits..." suggests that this study "proves" and "establishes".  These statements may not accurately reflect the research presented in this manuscript.
  3. Language issues with less than accurate definitions. For example, "Binge drinking is a behavioural pattern used to become drunk quickly" may not be the correct. 

Author Response

Reviewer 2:

Thank you for your work on this study.  I read the manuscript with interest and enthusiasm.  I especially appreciate the study design and analysis.  I would suggest the authors consider the following:

General reply to Reviewer 2:

We wish to thank you all for your constructive comments in this round of review. Your comments provided valuable insights to refine its contents and analysis. In this document, we try to address the issues raised as best as possible.

Comment 1:

The Aim is stated as to a. assess the attitude towards LM b. assess the attitude towards healthy lifestyle behaviors c. demonstrate an association with psychological wellbeing. It is unclear to me that these three Aims were clearly discussed in the paper and conclusion.  It is also unclear if these are the three items that the paper purports to address.

Reply 1:

The reviewer is correct to detect three aims of this study. However, the second aim is a bit different as we propose to assess the difference between Health and non-Health students towards healthy lifestyle behaviours. In the Discussions, we addressed the aim to demonstrate the association between virtuous lifestyle and psychological wellbeing at the first two indents. Then, we assessed the attitude towards LM at the third and fourth indents of the discussions, and we addressed the difference between Health and non-Health students at the fifth indent.

Comment 2:

There is language in the paper that are sweeping generalizations and unsupported and are not generalizable.  For example, the paragraph on page 8 that starts "Beyond these limits..." suggests that this study "proves" and "establishes".  These statements may not accurately reflect the research presented in this manuscript.

Reply 2:

We used weaker verbs as suggests, proposes and found.

Comment 3:

Language issues with less than accurate definitions. For example, "Binge drinking is a behavioural pattern used to become drunk quickly" may not be the correct.

Reply to comment 3:

We deleted the sentence “Binge drinking is a behavioural pattern used to become drunk quickly”
